# Polyphenolic Compounds Inhibit Osteoclast Differentiation While Reducing Autophagy through Limiting ROS and the Mitochondrial Membrane Potential

**DOI:** 10.3390/biom12091220

**Published:** 2022-09-01

**Authors:** Dipranjan Laha, Jaganmay Sarkar, Jyotirindra Maity, Asmita Pramanik, Md Sariful Islam Howlader, Derek Barthels, Hiranmoy Das

**Affiliations:** Department of Pharmaceutical Sciences, Jerry H. Hodge School of Pharmacy, Texas Tech University Health Sciences Center, Amarillo, TX 79106, USA

**Keywords:** autophagy, ellagic acid, gallic acid, mitochondrial membrane potential, osteoclast differentiation, ROS, tannic acid

## Abstract

Polyphenolic compounds are a diverse group of natural compounds that interact with various cellular proteins responsible for cell survival, differentiation, and apoptosis. However, it is yet to be established how these compounds interact in myeloid cells during their differentiation and the molecular and intracellular mechanisms involved. Osteoclasts are multinucleated cells that originate from myeloid cells. They resorb cartilage and bone, maintain bone homeostasis, and can cause pathogenesis. Autophagy is a cellular mechanism that is responsible for the degradation of damaged proteins and organelles within cells and helps maintain intracellular homeostasis. Imbalances in autophagy cause various pathological disorders. The current study investigated the role of several polyphenolic compounds, including tannic acid (TA), gallic acid (GA), and ellagic acid (EA) in the regulation of osteoclast differentiation of myeloid cells. We demonstrated that polyphenolic compounds inhibit osteoclast differentiation in a dose-dependent manner. Quantitative real-time PCR, immunocytochemistry, and western blotting revealed that osteoclast markers, such as NFATc1, Cathepsin K, and TRAP were inhibited after the addition of polyphenolic compounds during osteoclast differentiation. In our investigation into the molecular mechanisms, we found that the addition of polyphenolic compounds reduced the number of autophagic vesicles and the levels of LC3B, BECN1, ATG5, and ATG7 molecules through the inactivation of Akt, thus inhibiting the autophagy process. In addition, we found that by decreasing intracellular calcium and decreasing ROS levels, along with decreasing mitochondrial membrane potential, polyphenolic compounds inhibit osteoclast differentiation. Together, this study provides evidence that polyphenolic compounds inhibit osteoclast differentiation by reducing ROS production, autophagy, intracellular Ca^2+^ level, and mitochondrial membrane potentials.

## 1. Introduction

Polyphenolic compounds are primarily isolated from plants and are involved in various biological processes in animal cells. They play a protective role against various pathological conditions, including neurodegenerative and cardiovascular diseases, cancer, and osteoporosis [1]. Among the different polyphenolic compounds, tannic acid (TA), gallic acid (GA) and ellagic acid (EA) have strong antimicrobial, antivirus, anti-inflammatory, and anti-cancer properties [2]. TA is a water-soluble polyphenol that can be derived from various types of fruits, vegetables, wine, and tea [3]. GA (3,4,5-trihydroxy benzoic acid) is a bioactive compound found in plants and foods such as white tea and witch hazel [4]. EA is naturally available in its bound form as glycoside derivatives or as ellagitannins in nuts such as walnuts, common fruits such as pomegranates and grapes, and berry fruits such as raspberries, strawberries, and blueberries [5]. It was shown that some of the polyphenolic compounds exert their beneficial effect through their antioxidant properties since they scavenge reactive oxygen species (ROS) generated within cells [6]. The interaction between the polyphenolic compounds and intracellular signaling cascades including phosphatidylinositol-4,5-bisphosphate 3-kinase (PI3K), protein kinase B(PKB)/Akt, tyrosine kinase, protein kinase C (PKC), and mitogen-activated protein kinases (MAPKs), primarily lead to anti-inflammatory, chemo-preventive and chemotherapeutic activities [7,8]. The mode of action of polyphenolic compounds varies depending upon concentration and the application within a biological system [9].

Establishing a correlation between the presence of polyphenols in a compound and the beneficial effects mediated by polyphenols is very difficult because of the limited understanding of their bioavailability; generally, the small intestine can absorb polyphenols in the form of aglycones, but many of them in their native form are esters, glycosides or polymers that cannot be absorbed by the gut [10]. These compounds need to be metabolized by intestinal enzymes or the gut microflora [11]. Multiple studies showed that there is some degree of correlation between intake of polyphenols and bone health [12,13,14]. This is mainly due to their antioxidant properties, as oxidative stress plays an important role in the pathogenesis of osteoporosis [15]. Besides their scavenging properties, polyphenols can influence bone metabolism by downregulating inflammatory mediators [16] such as cytokines that are primarily implicated in sustaining osteoclast differentiation and activity [17,18]. This action contributes to a reduction in bone resorption. However, it is yet to be established how these polyphenolic compounds regulate myeloid cells during their osteoclast differentiation, and how intracellular and molecular mechanisms operate during this process.

Myeloid cells are an important subset of immune cells that play a critical regulatory role in mediating inflammation and destroying bone and cartilage in various skeletal diseases. Most bone and cartilage-related diseases are very complex in nature, where infiltration of myeloid cells takes place into the joint followed by interactions with local cellular constituents such as synovial fibroblasts and other infiltrated lymphocytes, leading to eventual tissue and cartilage destruction [19,20,21,22,23,24,25]. As myeloid cells are critical in tissue destruction processes, a greater understanding of the mechanisms of the monocyte differentiation process is important to specifically target those mechanisms. Osteoclasts are multinucleated cells, which are differentiated from myeloid cells and highly express tartrate-resistant acid phosphatase and calcitonin receptors. These cells are capable of resorbing bone and maintaining healthy bone turnover; however, hyper-activation of these cells leads to bone pathogenesis due to excessive bone destruction [26,27]. The complex process of osteoclast differentiation mainly depends on the rearrangements of the cytoskeletal structure and changes that occur in the organelles within the myeloid cells. It is shown that an intracellular protective mechanism, autophagy, is involved in these orchestrated processes of osteoclast differentiation that maintain bone homeostasis and develop pathogenic conditions in bone-related diseases through an unclear mechanism [28]. However, the activation of autophagy may scavenge reactive oxygen species, which contributes to increased osteogenic differentiation of bone marrow stem cells, showing the involvement of autophagy in bone biology [29].

A mammalian cell uses autophagy to recycle its damaged proteins and organelles through an intracellular catabolic process [30]. Generally, when cells are under stress or in adverse conditions, the autophagy process helps them to adapt to the harsh environmental conditions and ensures the availability of compounds necessary for the synthesis of required macromolecules. This process is key for cellular survival; however, long-term autophagy can lead to cell death [31]. Too much accumulation of damaged proteins and micro- and macro-organelles causes ROS production within the cells, which is ultimately deleterious for cells and the organism [31]. During various types of cellular differentiation, autophagy plays an important role [32]. However, impairment of the autophagy process leads to numerous pathological conditions, including rheumatoid arthritis and Paget’s disease in bone [33,34]. Our current study focuses on delineating and defining the molecular and cellular mechanisms involved in the osteoclast differentiation process in the presence of polyphenolic compounds. Establishing molecular and cellular mechanisms will facilitate the development of potential targeted therapeutics for the management of bone-related diseases.

## 2. Materials and Methods

### 2.1. Reagents and Antibodies

The monocytic cell line (RAW264.7) used for experiments was obtained from American Type Culture Collection, ATCC TIB-71. Soluble (s) RANKL (315-11-100UG) and macrophage colony-stimulating factor (M-CSF) (315-02-100UG) were obtained from Pepro Tech Incorporated. The tartrate-resistant acid phosphatase (TRAP) Assay Kit (387A-1KT), Tannic acid (1401-55-4), Gallic acid (149-91-7), and Ellagic acid (476-66-4), DCFDA (D6883), fluo 4 (F14201) were obtained from Sigma-Aldrich Corporation. DAPI (P36931) was purchased from Invitrogen Corporation, Waltham, MA, USA. Antibodies for BECN1 (3495S), ATG5 (12994S), LC3B (12741S), NFATc1 (8032), and GAPDH (2118S) were purchased from Cell Signaling Technology. Cathepsin K (ab188604) and TRAP (ab185716) antibodies were purchased from Abcam Inc. JC-1 Dye (T3168) and mitoSOX red compounds (M36008) were the product of Thermo Fischer Scientific, Waltham, MA, USA.

### 2.2. Osteoclast Differentiation

RAW264.7 cells were cultured in Dulbecco’s modified Eagle’s medium (DMEM) supplemented with 10% fetal bovine serum (FBS) and 100 U ml^−1^ penicillin-streptomycin all from GIBCO (Thermo Fisher Scientific, Waltham, MA, USA) and were incubated in 5% CO_2_ at 37 °C. Cells were harvested and sub-cultured according to experimental requirements. We have induced osteoclast differentiation from RAW264.7 cells by culturing in DMEM containing 10% heat-inactivated FBS in the presence of 20 ng/ml M-CSF and 40 ng/mL sRANKL for 6 days. In separate experiments, cells were also differentiated in the absence or presence of various non-toxic concentrations (0, 10, 20, 30, or 50 μM) of tannic acid, gallic acid, or ellagic acid. Fresh medium was replaced every two days in the presence or absence of stimulators and/or inhibitors.

### 2.3. TRAP Staining

To determine the presence of osteoclast differentiation, we stained cells with a TRAP staining kit following the manufacturer’s protocol. In brief, monocytes were cultured on coverslips placed in a 6-well plate for osteoclast differentiation with M-CSF and SRANKL in the presence or absence of TA, GA, or EA. On day 6 of the culture, each coverslip was removed from the wells, and cells were fixed with 4% paraformaldehyde in PBS for 20 min at room temperature and before being washed with PBS. Next, 5 μL of sodium nitrite solution and 5 μL Fast Garnet GBC base solution were mixed for each coverslip and incubated for 30 s. Then, 5 μl of acetate solution, 10 μl of naphthol AS-BI phosphate solution, 20 μl of tartrate solution, and 450 μl of deionized water (pre-warmed to 37 °C) were mixed. This solution was added onto each coverslip, which was placed into a well of a new 6-well plate and incubated for 1 h at 37 °C in a water bath in absence of light. Finally, the coverslips were rinsed with deionized water thoroughly, mounted on a glass slide, and examined under a light microscope, (Olympus Corporation of the Americas, Waltham, MA, USA, ix81). TRAP-positive cells (purple) containing at least three nuclei were counted as osteoclast cells.

### 2.4. Acidic Vesicular Organelles Staining

To determine the effect of polyphenolic compounds on autophagic vesicle formation during osteoclast differentiation, RAW264.7 cells were differentiated with M-CSF and sRANKL in the absence or presence of TA (30 μM), GA (30 μM), or EA (30 μM). Every 2 days, fresh medium with osteoclast differentiation stimulators and polyphenolic compounds was added to the cultures. On day 6, after the osteoclast differentiation, cells were stained with the auto-fluorescent compound monodansylcadaverine (MDC) according to the manufacturer’s protocol. In, brief, cells were incubated with a 50 mmol/L concentration of MDC at 37 °C for 15 min and washed with PBS. The cells were then mounted on a glass slide, viewed under a fluorescence microscope (Olympus Corporation of the Americas, Waltham, MA, USA, ix81), and images were captured digitally.

### 2.5. Western Blot Analysis

Western blot was performed to determine the levels of osteoclast differentiation-related marker proteins such as NFATc1, Cathepsin K, and TRAP, and the autophagic proteins BECN1, ATG5, ATG7, and LC3B in RAW264.7 cells during differentiation in the presence or absence of TA (30 μM), GA (30 μM), or EA (30 μM). GAPDH was used as an internal control. The cells were lysed in 100 μL pre-cooled RIPA lysis buffer (Millipore, Sigma Aldrich Corporation, St. Louis, MO, USA, #20-188, 20-188) for 30 min on ice and centrifuged at 12,000× *g* for 12 min. The supernatant was collected and the protein concentration was estimated with Bradford’s reagent (Bio-Rad Laboratories, Hercules, CA, USA, #5000006), using bovine serum albumin (BSA) (Sigma Aldrich Corporation, St. Louis, MO, USA, #A7906-10G) as a standard. Equal amounts of protein (40 μg) were separated by SDS-PAGE gels electrophoretically and transferred to a polyvinylidene difluoride membrane (Bio-Rad Laboratories, Hercules, CA, USA, #1620115). After blocking with 5% BSA for 1 h at room temperature, the membranes were probed with primary antibodies for 12–16 h at 4 °C. Then, membranes were incubated with an appropriate HRP (horseradish peroxidase)-labeled secondary antibodies (Cell Signaling Technology, Danvers, MA, USA, #7074, and #7076) for 2 h at room temperature. Immunoreactive protein bands were visualized by enhanced chemiluminescence (ECL, Amersham Pharmacia Biotechnology, Amersham, UK, #RPN2232), and the band detections were kept within the linear range.

### 2.6. RNA Extraction and Real-Time PCR

TRIzol reagent (Invitrogen Corporation, Waltham, MA, USA, #15596026) was used for the extraction of RNA, and cDNA was prepared using the PrimeScript reverse transcription kit according to the manufacturer’s protocols. Real-time PCR amplification reactions were performed with the SYBR Green PCR Kit (Applied Biosystem, Waltham, MA, USA, #4309155). The relative expression of each target gene was quantified by calculating 2^−ΔΔCT^ (threshold cycle) values normalized by β-Actin levels. Each sample was analyzed in triplicate. The following primer sets were purchased from Sigma Aldrich Corporation and were used for amplification. Becn1, 5′-TTT TCT GGA CTG TGT GCA GC-3′ (forward), 3′ -GCTTTTGTCCACTGCTCCTC-5′ (reverse); Map1lc3b, 5′-GTCAGATCGTCTGGCTCGG-3′ (forward), 5′-TGCAAGCGCCGTCTGATTAT-3′ (reverse); Atg7, 5′-TTGAGCGGCGACAGCATTA3′ (forward), 5′-TGAGGAAAGCCTCATGGCAG-3′(reverse); β-Actin 5′-GGCACCACACCTTCTACAATG-3′ (forward), 5′-GGGGTGTTGAAGGTCTCAAAC-3′ (reverse); Ctsk5′-ATGTGAACCATGCAGTGTTGGTGG-3′ (forward)’ 5′-ATGCCGCAGGCGTTGTTCTTATTC-3′(reverse); Nfatc1 5′-AGATGGTGCTGTCTGGCCATAACT-3′(forward), 5′-TGCGGAAAGGTGGTATCTCAACAA-3′ (reverse). Trap 5′-GCCTTGTCAAGAACTTGCGACCAT-3′(forward), 5′-TTCGTTGATGTCGCACAGAGGGAT -3′ (reverse).

### 2.7. Immunofluorescence Staining

To investigate osteoclast differentiation in the absence or presence of TA (30 μM), GA (30 μM), or EA (30 μM), immunofluorescence analysis was performed for NFATC1, Cathepsin K, and TRAP molecules. In brief, RAW264.7 cells were grown on sterile coverslips inserted into a 6-well plate. After 24 h of culture, cells underwent osteoclast differentiation with M-CSF and sRANKL in the absence or presence of TA, GA, or EA compounds. After 6 days of differentiation, cells were fixed with 4% paraformaldehyde (Santa Cruz Biotechnology, Dallas, TX, USA, #sc-281692) for 30 min. After washing with PBS, cells were permeabilized with 0.1% Triton X-100 (Sigma Aldrich Corporation, St. Louis, MO, USA, #T8787) and blocked with 1% BSA. Then, cells were incubated with 200 μL of primary antibody (1:200) overnight at 4 °C. After washing with PBS, cells were incubated with 200 μL of secondary anti-rabbit or anti-mouse antibodies (Alexa Fluor 488, A11001 or Alexa Fluor 594, A21235; 1:2000 dilution; Invitrogen Corporation) for 45 min in the dark. After incubation, cells were washed thrice with PBS (GIBCO, Thermo Fisher Scientific, Waltham, MA, USA, #70013-032) and mounted with 4, 6-diamidino-2-phenylindole, dihydrochloride (DAPI, Invitrogen Corporation, Waltham, MA, USA, #D1306) on glass slides and sealed with transparent nail varnish. Slides were viewed under a fluorescence microscope and images were captured digitally using an Olympus ix81 microscope with Slidebook 5.0 × 64 software (Olympus, Tokyo, Japan).

### 2.8. Detection of Mitochondrial Membrane Potential

To determine mitochondrial membrane potential after the addition of TA (30 μM), GA (30 μM), or EA (30 μM) compounds during 6 days of osteoclast differentiation of RAW264.7 cells, staining was performed using JC-1 dye. In brief, RAW264.7 cells were grown on sterile coverslips inserted into each well of a 6-well plate. After 24 h of culture, cells underwent osteoclast differentiation with M-CSF and sRANKL in the absence or presence of TA, GA, or EA compounds. After 6 days of differentiation, cells were stained with JC-1 dye, mounted on a slide with DAPI, and sealed with transparent nail varnish. Imaging was performed under a super-resolution confocal microscope (Leica Stellaris 8 STED, Wetzlar, Germany) using a 100x objective, and images were analyzed using LAS X image analysis software. Each experiment was performed in triplicate, and experiments were performed at least three times.

### 2.9. Detection of ROS

Reactive oxygen species (ROS) detection was performed by using the fluorogenic dye 2′,7′- dichlorodihydrofluorescein diacetate (DCFDA), which enters the cells and interacts with a reactive oxygen molecule to form the green fluorescent compound dichlorodihydrofluorescein (DCF). In short, a stock solution of DCFDA (10 mM) was prepared in methanol and was further diluted with culture medium to a working solution of 100 µM. RAW264.7 cells (2 × 10^5^) were seeded on a coverslip inserted into each well of a 6-well plate and incubated overnight. The next day, cells were subjected to osteoclast differentiation in the presence or absence of each of the polyphenolic compounds (30 μM). After treatments, coverslips were washed with ice-cold Hank’s balanced salt solution (HBSS) and incubated with 100 µM of DCFDA for 30 min at 37 °C. After washing with 1 × PBS, the coverslips were mounted on glass slides with DAPI. Imaging was performed under a super-resolution confocal microscope (Leica Stellaris 8 STED, Wetzlar, Germany) using a 100× objective, and images were analyzed using LAS X image analysis software. Each experiment was performed in triplicate, and experiments were performed at least three times.

### 2.10. Detection of Mitochondrial ROS

Cellular mitochondrial ROS generation was evaluated by using the mitoSOX red compound (#M36008, Thermo Fisher Scientific, Waltham, MA, USA). Superoxide compounds present in mitochondria oxidize mitoSOX red to produce red fluorescence. In short, we prepared a 5 mM stock solution of mitoSOX in DMSO. RAW264.7 cells (2 × 10^5^) were seeded on a coverslip inserted into each well of a 6-well plate and kept overnight. The next day, cells were subjected to osteoclast differentiation in the presence or absence of each of the polyphenolic compounds (30 μM). Coverslips were then washed with ice-cold 1 x PBS and incubated in a 2 µM working solution of the mitoSOX red for 30 min at 37 °C. After washing with 1 × PBS, the coverslips were mounted on glass slides with DAPI. Imaging was performed under a super-resolution confocal microscope (Leica Stellaris 8 STED, Wetzlar, Germany) using a 100× objective, and images were analyzed using LAS X image analysis software. Each experiment was performed in triplicate, and experiments were performed at least three times.

### 2.11. Intercellular Ca^2+^ Measurement

Intracellular Ca^2+^ was measured using a Calcium Assay Kit (#ab102505, Abcam, Cambridge, MA, USA) according to the manufacturer’s protocol. In brief, after 6-days of osteoclast differentiation with M-CSF and sRANKL in the absence or presence of TA, GA, or EA (30 μM), cells were harvested and homogenized with calcium assay buffer. 50 µL of cell lysate was added to each well of a 96-wells plate. 90 µL of the Chromogenic Reagent was added to each well, followed by 60 µL of Calcium Assay Buffer. Samples were mixed and incubated at RT for 10 min in the dark. A standard curve was prepared using the standard dilutions. Absorbance was recorded at 575 nm using a microplate reader (Synergy 2, BioTek Instruments, Inc., Winooski, VT, USA). Each experiment was performed in triplicate, and experiments were performed at least three times.

### 2.12. Cell Viability Assay

To determine the effect of EA, GA, and TA on cell viability, MTT assays were performed. In short, RAW264.7 cells (2 × 10^3^/well) were seeded in the 96-well plate and incubated for 24 h at 37 °C with 5% CO_2_. The next day, the media was replaced with or without varying concentrations (5, 10, 20, 30, 50, 100 µM) of EA, GA, and TA, and incubated for 24 h and 48 h. Then, cell viability was analyzed via MTT assay using a Roche Cell Proliferation Kit I (Roche Diagnostics, Basel, Switzerland; #11465007001) according to the manufacturer’s protocol. The absorbance was measured using a microplate reader (Synergy 2, BioTeK Instruments Inc, Winooski, VT, USA). For each experiment, the results were expressed as the percentage viability according to the following formula:% Viability = 100 × (absorbance of treatment/absorbance of control).

### 2.13. Statistical Analysis

All experiments were performed at least three times in triplicate, and the results were displayed as mean ± SEM. Statistical analyses were performed using GraphPad Prism 5.0 for Windows (GraphPad Software, San Diego, CA, USA). In the statistical analysis, a one-way ANOVA was used with Dunnett’s multiple-comparison test.

## 3. Results

### 3.1. Effect of Polyphenolic Compounds on Osteoclast Differentiation

To determine the effects of tannic acid (TA), gallic acid (GA), and ellagic acid (EA) on the osteoclast differentiation of myeloid cells (Appendix A), RAW264.7 cells were induced for osteoclast differentiation with M-CSF and sRANKL in absence or presence of various concentrations of TA, GA, or EA molecules. Tartrate-resistant acid phosphatase (TRAP) staining after 6 days of differentiation revealed a remarkable dose-dependent reduction in the number of TRAP-positive cells in plates containing any of the polyphenolic compounds (Figure 1A). When TRAP-positive cells were counted under a microscope, we found that a significantly reduced number of osteoclasts were present in samples that underwent differentiation in the presence of polyphenolic compounds compared to their respective controls. However, it seems TA was more effective than GA or EA in reducing osteoclast differentiation (Figure 1B). Dose-dependent effect of ellagic acid (EA), gallic acid (GA), and tannic acid (TA) on RAW264.7 cell viability was also determined by MTT assay. We did not notice any remarkable cell death using the 30 μM concentration of EA, GA, or TA (Appendix A).

### 3.2. Effect of Polyphenolic Compounds on Actin Ring Formation

The bone resorption process is dependent on the dynamics of actin cytoskeleton structures of the osteoclasts [35]. Therefore, we have determined the effect of EA, GA, and TA on the formation of actin ring cytoskeletons during osteoclastic differentiation. The actin rings were visualized after the staining with FITC-conjugated phalloidin. Our confocal microscopy images showed that in mature osteoclasts, F-actin was arranged into a ring-like structure in the periphery of the differentiated cells and that the polyphenolic compounds mediated the shrinkage of osteoclasts and disrupted the formation of actin ring structures (Appendix A).

### 3.3. Effect of Polyphenolic Compounds on Osteoclast-Related Gene Expressions during Differentiation

To determine the effect of polyphenolic compounds on osteoclast differentiation-related molecular marker expressions in myeloid cells, cells were harvested after 6 days of differentiation and total RNA was isolated. Quantitative RT-PCR was performed to determine the genetic expressions of osteoclast differentiation-related markers, such as TRAP, cathepsin K (Ctsk), and NFATc1. We found significantly reduced levels in expressions of TRAP, cathepsin K, and NFATc1 in cells that were differentiated in the presence of TA, GA, or EA compared to respective controls (Figure 2A).

### 3.4. Effect of TA, GA, and EA on Protein Levels of Osteoclast Differentiation Marker Molecules during Osteoclastogenesis

To evaluate whether polyphenolic compounds exert any effect on osteoclast differentiation-related molecular levels in myeloid cells, cells differentiated for 6 days were subjected to immunocytochemical staining for NFATc1, cathepsin K and TRAP. Immunocytochemical staining showed reduced levels of expression of osteoclast differentiation-related molecules in cells where any of the polyphenolic compounds were present during differentiation (Figure 2B). We further confirmed protein levels of the osteoclast differentiation-related molecules by western blotting methods and found that NFATc1, cathepsin K, and TRAP levels were indeed decreased in differentiated cells that were cultured in presence of TA, GA, or EA compared to respective controls (Figure 4A and Appendix A).

### 3.5. Effect of Polyphenolic Compounds on the Expression of Autophagic Genes in Myeloid Cells during Osteoclast Differentiation

To determine the molecular and cellular mechanisms involved in the inhibition of osteoclast differentiation by polyphenolic compounds, we have investigated the expression of autophagy-related genes after 6 days of differentiation in the absence or presence of the polyphenolic compounds. Quantitative RT-PCR was performed from isolated RNAs to determine the genetic expressions of autophagic molecular markers, such as Beclin1, LC3B, and ATG7. We found significantly reduced levels in expressions of Beclin1, LC3B, and ATG7 in cells that were subjected to differentiation in presence of any of the polyphenolic compounds compared to respective controls (Figure 3A). To investigate autophagic vesicle formation in cells, MDC staining was performed. We found that autophagosomes were recruited during the osteoclast differentiation of myeloid cells. However, recruitment of autophagosomes were not prominent after the addition of TA, GA, or EA (Figure 3B). We further confirmed protein levels of the autophagy-related molecules by western blotting and found that Beclin1, LC3B, ATG5, and ATG7 levels were indeed decreased in differentiated cells which were cultured in presence of TA, GA, or EA compared to respective controls (Figure 4B and Appendix A).

### 3.6. Effect of Polyphenolic Compounds on the Expression of Apoptotic and Cell Survival-Related Molecules in Myeloid Cells during Osteoclast Differentiation

To determine the apoptosis in myeloid cells after osteoclast differentiation with the addition of polyphenolic compounds, western blots were performed after 6 days of differentiation. Western blot data revealed that apoptotic proteins PARP and BAX were decreased or unaltered during the osteoclast differentiation, respectively, and the addition of TA, GA, or EA remarkably increased the levels of PARP, cleaved PARP, and BAX molecules. However, the cell survival gene Bcl2 was decreased (Figure 4C and Appendix A).

### 3.7. Determination of Downstream Signaling Pathways

Further studies were performed to determine the downstream activation of relevant molecules during osteoclast differentiation. Our study revealed that the activation of Akt was remarkably reduced with little alteration in the activation of p38 after the addition of TA, GA, or EA compounds during the osteoclast differentiation. There was a remarkable reduction of total Erk1/2 levels after the addition of the polyphenolic compounds tested. Together, these findings provide evidence that the activation of Akt might have played a critical role in inhibiting autophagy and resulted in the induction of apoptosis in presence of TA, GA, or EA compounds (Figure 4D and Appendix A) which is in agreement with the previous report [34].

### 3.8. Effect of Polyphenolic Compounds on Production of ROS, and Intracellular Ca^2+^ during Osteoclast Differentiation

It is already reported that ROS production generates oxidative stress and promotes osteoclast differentiation and maturation. We further studied the status of reactive oxygen species (ROS) generation during the osteoclast differentiation in the presence or absence of polyphenolic compounds using immunostaining with DCFDA and mitoSOX. We observed that the level of DCFDA and mitoSOX staining was appeared to be increased after the differentiation of osteoclasts. However, after the addition of polyphenolic compounds the level of ROS as determined by DCFDA staining (Figure 5) and mitoSOX staining (Figure 6), was appeared to be decreased during differentiation.

Calcium plays an important role in bone remodeling. Both extracellular and intracellular Ca^2+^ play crucial roles that are involved in the regulation of cell proliferation and differentiation. It was shown that the elevation of intercellular Ca^2+^ is due to a higher release of Ca^2+^ from the endoplasmic reticulum and a simultaneous influx of Ca^2+^ via voltage-gated Ca^2+^ channels [36]. We also observed that the level of intracellular Ca^2+^ was significantly increased in osteoclasts; however, after the addition of each of the polyphenolic compounds, the level of Ca^2+^ was significantly decreased (Figure 7).

### 3.9. Effect of Polyphenolic Compounds on Mitochondrial Membrane Potential during Osteoclast Differentiation

To determine the mitochondrial membrane potential, we assessed osteoclast differentiated cells after 6 days of differentiation in the absence or presence of polyphenolic compounds after staining with JC-1 dye. Mitochondrial membrane polarization is specifically indicated by staining with the cationic JC-1 dye. This dye exhibits potential-dependent accumulation in mitochondria, indicated by a fluorescence emission shift from green to red as the it accumulates within mitochondrial membranes. It was observed that TA-, GA-, or EA-treated cells show mostly green fluorescent intensity (indicating high amounts of the monomer-form of the dye and few intact mitochondrial membranes) compared to the untreated cells, which appeared redder (indicating the aggregated-form of the dye in intact mitochondria) (Figure 8). These results confirmed that polyphenolic compounds reduced mitochondrial membrane integrity, thereby reducing the mitochondrial membrane potentials.

## 4. Discussion

Oxidative damage due to redox imbalance plays a critical role in several metabolic changes that subsequently develop disease states; therefore, developing targeted therapies for neutralizing oxidative stress has taken high priority in attempts to treat those disease states. As bioactive polyphenolic compounds provide enormous health benefits with their antioxidant and anti-inflammatory properties [37], a new field of interest has emerged involving osteo-immune-oncology, bone metabolism, and tumorigenesis [5,6,9,12,38]. The data we have collected is in agreement with much of the previous literature on the interactions between polyphenolic compounds and osteoclastogenesis [39,40,41,42,43,44,45,46]. Antioxidant properties are well-studied for EA, GA, and TA [37,47,48], which is why we have chosen those molecules for the immune modulation of monocytes in our current study. It was shown that paraquat (PQ, a ROS-inducing compound) elevates ROS production and that EA significantly reduces intracellular ROS levels through the Nrf2 (nuclear factor-erythroid 2-related factor) pathway in A549 cells (lung carcinoma cells) [49]. It is also reported that TA inhibits oxidative stress, inflammation, and apoptosis via the NF-κB/Nrf2 signaling axis in arsenic trioxide (ATO)-induced nephrotoxicity [48]. In a separate study, it was shown that administration of GA improved the lethal effect of PQ-induced oxidative stress on renal tissue in a murine model [50].

Herein, we focused on identifying the role of polyphenolic compounds during osteoclast differentiation [34,51]. Osteoclast differentiation and bone resorption require a series of regulatory molecules and hydrolytic enzymes to maintain dynamic homeostasis with osteoblasts. There are almost 8000 recognized polyphenols identified, of which approximately 500 are biologically active. In the current study, we have selected three polyphenolic compounds of different natures, including TA, GA, and EA, to investigate their role in the regulation of osteoclast differentiation of myeloid cells. First, we differentiated RAW264.7 (myeloid) cells to osteoclasts in the presence or absence of poly phenolic compounds and performed TRAP staining to confirm osteoclast differentiation. In addition, several osteoclast-specific markers such as NFATc1, Cathepsin K, and TRAP were evaluated for their gene expression and protein expressions using qRT-PCR, western blot, and immunofluorescence staining, respectively. TRAP staining, qRT-PCR, western blot, and immunofluorescence staining showed that osteoclast differentiation was remarkably reduced in the presence of any of the polyphenolic compounds tested. One of the most-studied bioactive compounds in recent years shows a heavy flavonoid component of Chinese tea that has been reported to have intriguing osteogenic properties [52], which is consistent with our results.

From our previous study and other reports, we found that autophagy plays a major role during osteoclast differentiation [34,51]. We hypothesized that TA, GA, and EA might regulate the osteoclastogenesis process by modulating autophagy and autophagosome formation. To test this, we analyzed autophagy-related molecules after the addition of polyphenolic compounds during osteoclast differentiation. We observed a significantly reduced level in expressions of Beclin1, LC3B, and ATG7 in cells that underwent differentiation in presence of TA, GA, or EA, both in mRNA and protein levels. We further found that recruitment of autophagosomes was not prominent after the addition of any of the polyphenolic compounds. These findings are in alignment with the previous reports in cancer cells showing the involvement of autophagy in reducing cancer burden [38]. To investigate apoptosis-related molecules, we found that the levels of BAX and PAPR were increased, and conversely, cell survival gene Bcl2 was decreased upon the addition of TA, GA, or EA during osteoclast differentiation. We also verified the activation of Akt and p38 molecules, which play important roles in autophagy and apoptosis during osteoclast differentiation. From our study, we found that activation of Akt is remarkably reduced upon the addition of any of the polyphenolic compounds tested. However, there was no significant change in the activation of the p38 molecule. Basal Erk1/2 levels were also remarkably reduced upon the addition of any of the polyphenolic compounds tested. Previous reports showed that Akt is a regulator of autophagy [53], which might play an important role in reducing osteoclast differentiation. Our data demonstrated that autophagy was induced during osteoclast differentiation and was inhibited by the polyphenolic compounds TA, GA, and EA.

However, how autophagy was regulated through polyphenolic compounds is yet to be defined. To test this, we next measured levels of various biochemical parameters like reactive oxygen species, mitochondrial superoxide, intracellular Ca^2+^, and mitochondrial membrane potential in the presence or absence of TA, GA, or EA during osteoclast differentiation. We found that after the addition of polyphenolic compounds during osteoclast differentiation, reactive oxygen species were significantly decreased and the intracellular Ca^2+^ levels were also decreased. Previous reports showed that Ca^2+^ signaling activates differentiation in osteoclast precursors but suppresses resorption in mature osteoclasts [54], which is consistent with our findings. Recent studies have also indicated that the level of ROS is important during the differentiation process of osteoclasts [55]. The Ca^2+^ signaling network involves many different ways to regulate cellular processes that function over a wide range of activities and functions, including through the maintenance of buffers, pumps, and exchangers on the plasma membrane, and in maintaining internal cellular stores. Calcium signaling pathways also interact with other cellular signaling pathways including those that involve the production of ROS [56]. Our data show a reduced amount of ROS after the addition of TA, GA, or EA during osteoclast differentiation, which might have helped to inhibit the differentiation of osteoclast cells by limiting damage to intracellular organelles. Several groups reported that mitochondrial superoxide production is a major cause of the cellular oxidative damage that may be the underlying cause of degradative diseases and aging [15]. Therefore, we also measured mitochondrial superoxide levels in the presence of TA, GA, or EA during osteoclast differentiation and found that mitochondrial superoxide generation (mitoSOX) is very similar to the ROS data. We believe that the inhibition of ROS and mitochondrial superoxide played a critical role in limiting damage to intracellular organelles, including mitochondrial membranes, in osteoclast cells thus inhibited the differentiation process. To confirm our hypothesis mitochondrial membrane potential was evaluated by JC-1 dye staining. The staining exhibited mitochondrial membrane depolarization after the addition of the TA, GA, or EA during osteoclast differentiation. These results confirmed that polyphenolic compounds induced damage to the mitochondrial membranes and thereby reduced the mitochondrial membrane potential, which ultimately caused the inhibition of osteoclast differentiation.

## 5. Conclusions

The present study demonstrates that EA, GA, and TA regulate osteoclast differentiation and osteoclast markers in myeloid cells, such as NFATc1, Cathepsin K, and TRAP. We have demonstrated that these polyphenolic compounds reduce the levels of autophagic vesicles and molecules indicated by BECN1, ATG5, ATG7, and LC3B. Moreover, we found that polyphenolic compounds inhibit osteoclast differentiation by decreasing levels of ROS determined by DCFDA and mitoSOX staining, and decreasing levels of intracellular calcium. These results, along with the decrease in ROS and mitochondrial membrane potential caused by TA, GA, and EA, provide evidence that these compounds can inhibit osteoclast differentiation.

## Figures and Tables

**Figure 1 biomolecules-12-01220-f001:**
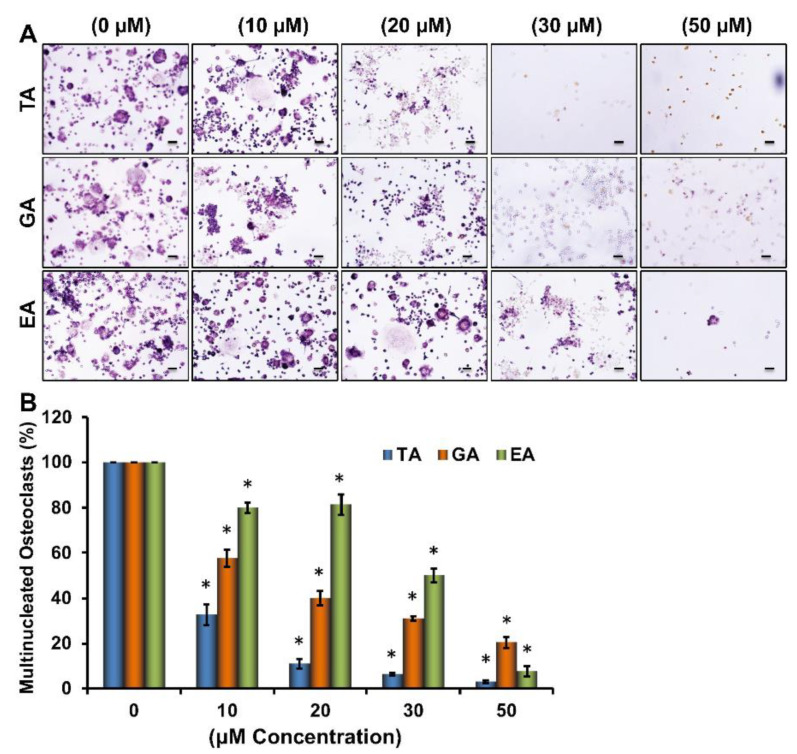
Dose-dependent inhibition of osteoclast differentiation by polyphenolic compounds. (**A**). Images of osteoclast differentiation as identified by TRAP staining at day 6 in the absence or presence of stated concentrations of tannic acid (TA), gallic acid (GA), and ellagic acid (EA) during osteoclast differentiation (Scale bar = 20 μm). (**B**). The number of TRAP-positive multinucleated osteoclasts present in each group shown graphically in the absence or presence of stated concentrations of stated polyphenolic compounds during osteoclast differentiation. * *p* < 0.05 indicates statistical significance between control vs. TA, control vs. GA, control vs. EA.

**Figure 2 biomolecules-12-01220-f002:**
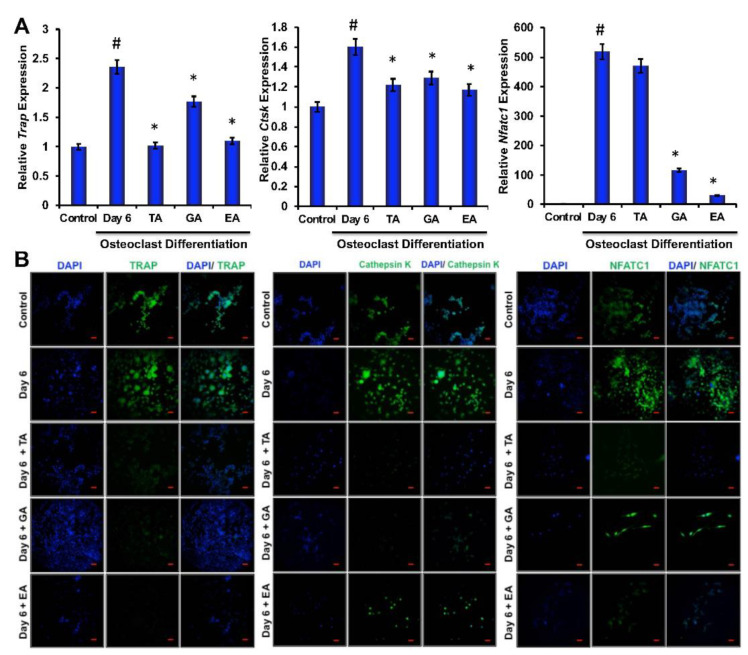
Polyphenolic compounds inhibit osteoclast differentiation-related molecules. (**A**). The quantitative real-time PCR analysis of osteoclast differentiation marker genes such as TRAP, Cathepsin K, and NFATc1 was shown graphically on day 6 in the absence or presence of tannic acid (TA), gallic acid (GA), or ellagic acid (EA) during osteoclast differentiation. (# indicates *p* < 0.05 when compared between control and day 6 of osteoclast differentiated samples, * indicates *p* < 0.05, when compared between day 6 of osteoclast differentiated samples in the absence or presence of either of the polyphenolic compounds) (**B**). Immunocytochemically stained images of osteoclast differentiation markers such as TRAP, Cathepsin K, or NFATc1 staining at day 6 in the absence or presence of tannic acid (TA), gallic acid (GA), or ellagic acid (EA) were shown (Scale bar = 20 μm).

**Figure 3 biomolecules-12-01220-f003:**
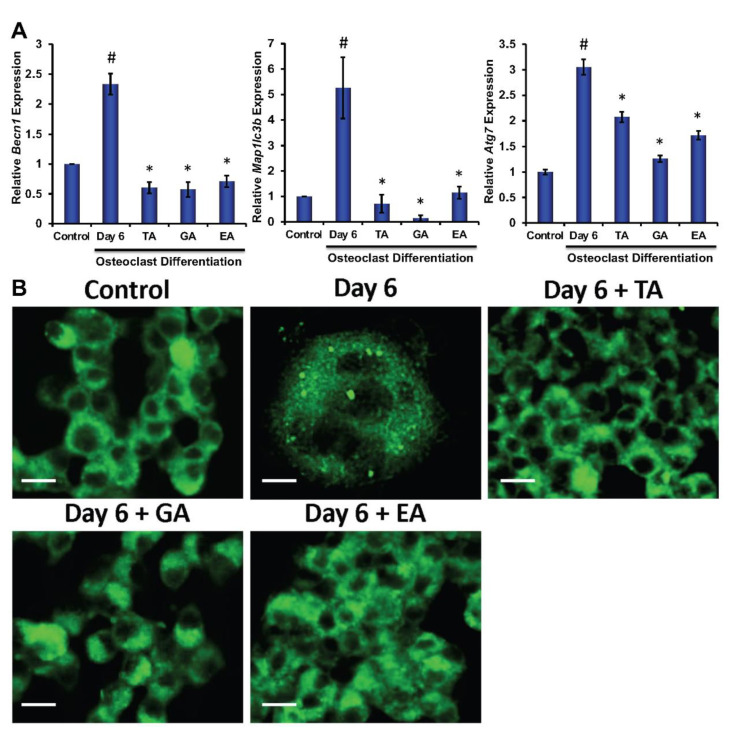
Polyphenolic compounds reduce autophagic gene expressions and autophagic vesicle formations. (**A**). The quantitative real-time PCR analysis of autophagy-related marker genes such as Beclin1 (*Becn1*), LC3B (*map1lcb*), and ATG7 (*Atg7*) was shown graphically at day 6 in the absence or presence of tannic acid (TA), gallic acid (GA), or ellagic acid (EA) during osteoclast differentiation. (# indicates *p* < 0.05 when compared between control and day 6 of osteoclast differentiated samples, * indicates *p* < 0.05, when compared between day 6 of osteoclast differentiated samples in the absence or presence of either of the polyphenolic compounds) (**B**). Images of the stained autophagic vesicles were shown after osteoclast differentiation in cells in the presence or absence of polyphenolic compounds during differentiation. Control indicates undifferentiated cells (Scale bar = 20 μm).

**Figure 4 biomolecules-12-01220-f004:**
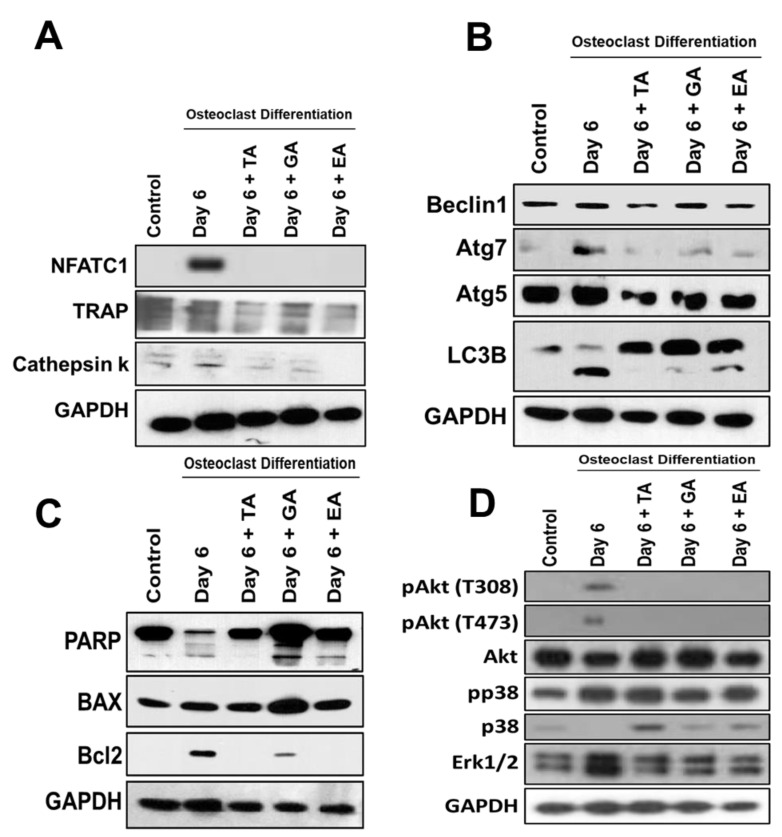
Reduced osteoclast differentiation- and autophagy-related proteins and induced apoptosis-related proteins after the addition of polyphenolic compounds. (**A**). Western blotting of osteoclast differentiation-related molecules such as NFATc1, TRAP, and Cathepsin K protein levels, with GAPDH used as an internal control, was shown at day 6 in the absence or presence of tannic acid (TA), gallic acid (GA), or ellagic acid (EA) during osteoclast differentiation. (**B**). Western blotting of autophagy-related molecules such as Beclin1, Atg7, Atg5, and LC3B protein levels, with GAPDH used as an internal control, was shown on day 6 in the absence or presence of tannic acid (TA), gallic acid (GA), or ellagic acid (EA) during osteoclast differentiation. (**C**). Western blotting of apoptosis and cell survival-related molecules such as PARP, BAX2, and Bcl2 protein levels, keeping GAPDH as an internal control, was shown on day 6 in the absence or presence of tannic acid (TA), gallic acid (GA), and ellagic acid (EA) during osteoclast differentiation. (**D**). Western blotting of down-stream signaling activation-related molecules such as pAkt (T308 and T473), pP38, total Akt, total p38, and total Erk1/2 protein levels, with GAPDH as an internal control, were shown at day 6 in the absence or presence of tannic acid (TA), gallic acid (GA), or ellagic acid (EA) during osteoclast differentiation.

**Figure 5 biomolecules-12-01220-f005:**
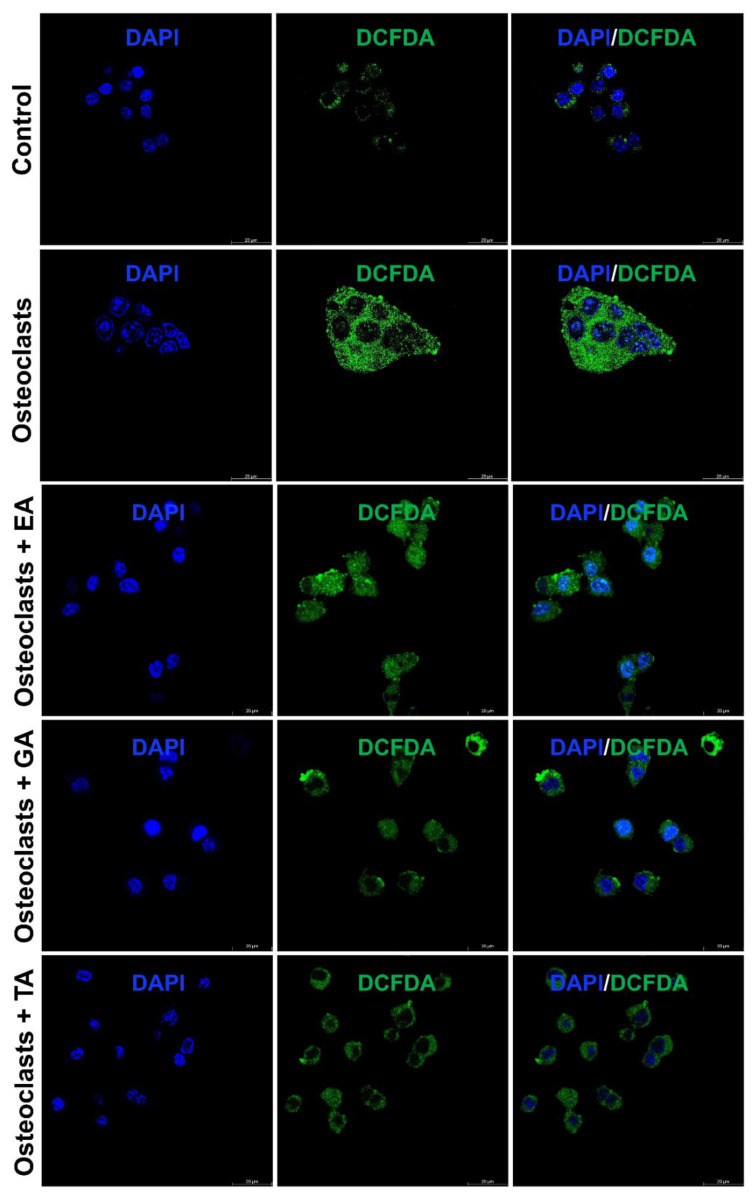
Polyphenolic compounds reduce ROS production during osteoclast differentiation. Evaluation of ROS production levels, determined by DCFDA fluorescence staining was shown at day 6 in the absence or presence of ellagic acid (EA), gallic acid (GA), or tannic acid (TA) during osteoclast differentiation.

**Figure 6 biomolecules-12-01220-f006:**
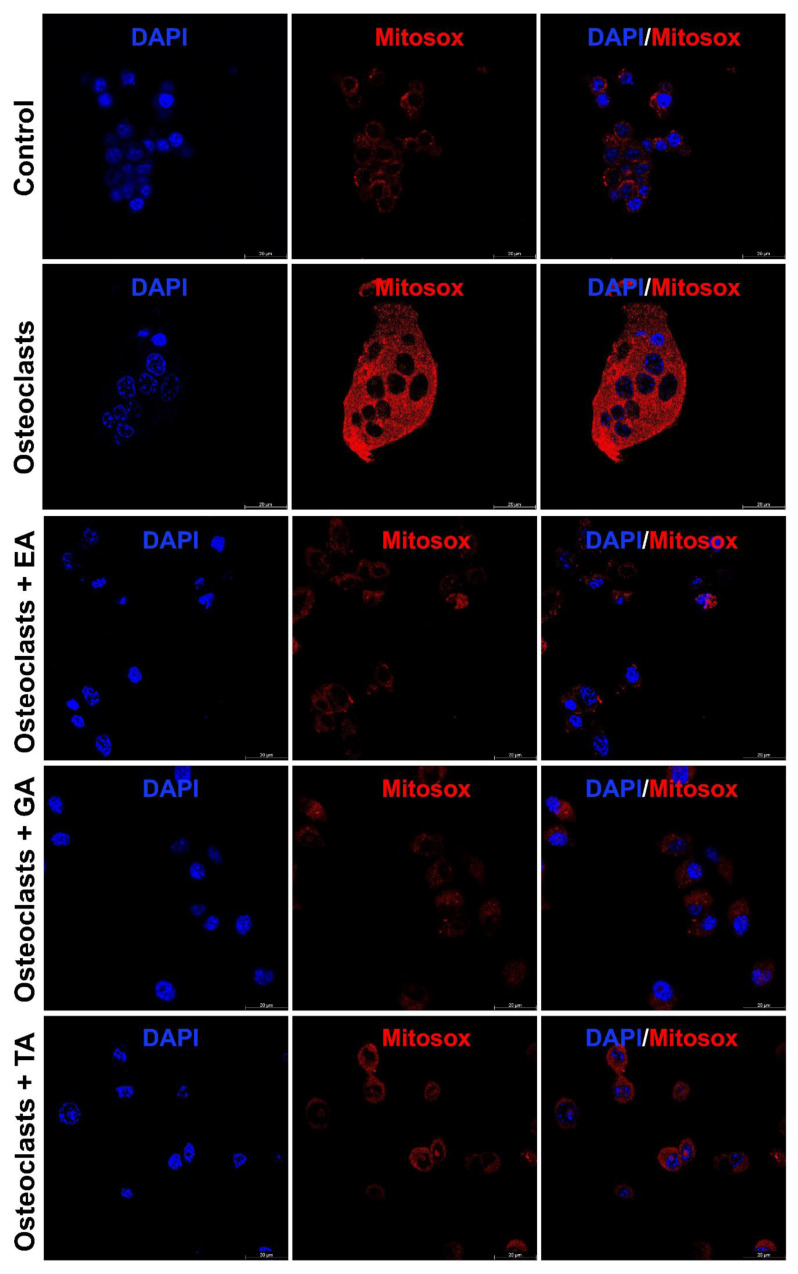
Polyphenolic compounds reduce mitochondrial superoxide production during osteoclast differentiation. Evaluation of mitochondrial superoxide production levels was shown by mitoSOX staining on day 6 in the absence or presence of ellagic acid (EA), gallic acid (GA), or tannic acid (TA) during osteoclast differentiation.

**Figure 7 biomolecules-12-01220-f007:**
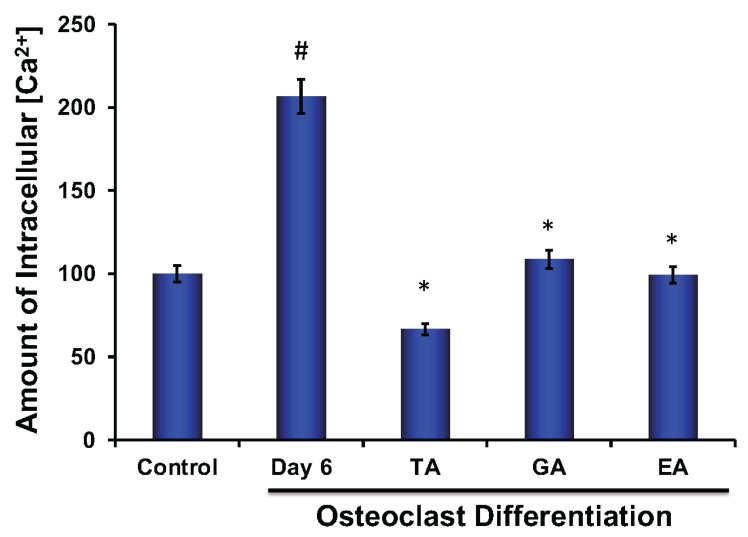
Polyphenolic compounds reduce intracellular Ca^2+^ levels during osteoclast differentiation. Evaluation of intracellular Ca^2+^ level was shown graphically at day 6 in the absence or presence of tannic acid (TA), gallic acid (GA), or ellagic acid (EA) during osteoclast differentiation. (# indicates *p* < 0.05 when compared between control and day 6 of osteoclast differentiated samples, * indicates *p* < 0.05, when compared between day 6 of osteoclast differentiated samples in the absence or presence of either of the polyphenolic compounds).

**Figure 8 biomolecules-12-01220-f008:**
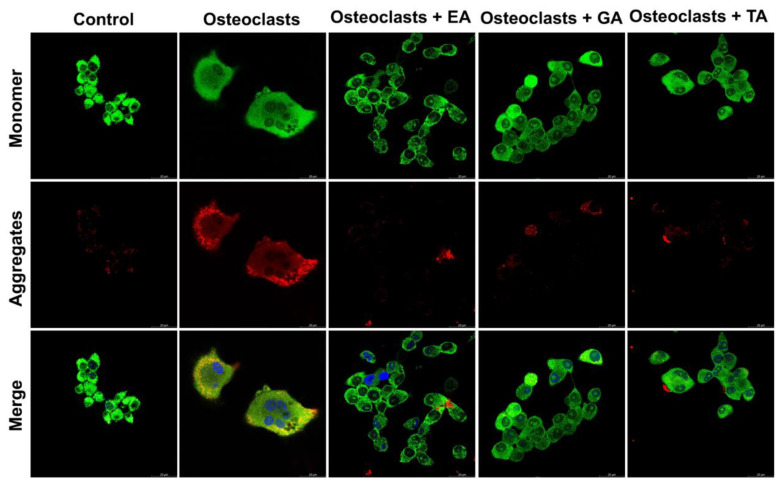
Polyphenolic compounds reduce mitochondrial membrane potential in differentiated osteoclasts. Images of stained mitochondrial membrane potential in differentiated osteoclasts are presented in the absence or presence of ellagic acid (EA), gallic acid (GA), and tannic acid (TA) keeping undifferentiated cells as a control. Green color indicates the monomeric form of the stained JC-1 dye, and red color indicates an aggregate form of the stained JC-1 dye.

## Data Availability

The datasets used and/or analyzed during the current study will be available on reasonable request.

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
