# Peer review of "Polyphenolic Compounds Inhibit Osteoclast Differentiation While Reducing Autophagy through Limiting ROS and the Mitochondrial Membrane Potential"

_biomolecules, 2022, doi:10.3390/biom12091220_

Round 1

Reviewer 1 Report

Comments to authors,

The manuscript has fatal errors.

Major points:

1. The evidence presented in the manuscript dose not sound new.

2. Not causal relationship but only parallelisms are presented. The authors have to present the evidence that RAW246.7 cells conduct autophagy and differentiate into osteoclast in the presence of the polyphenolic compounds when ROS are produced, and that RAW246.7 cells conduct autophagy and differentiate into osteoclast in the presence of the polyphenolic compounds when mitochondria are depolarized, to insist that the polyphenolic compounds prevent RAW246.71 cells from differentiating into osteoclasts by suppressing autophagy through loss of mitochondrial membrane potential and suppression of ROS production, as the authors titled the manuscript.

3. The cultured cells used in the study are probably under bad conditions, making the obtained results not-reliable. Perhaps, the concentrations of the polyphenolic compounds used was high and cytotoxic.

4. Citation is not appropriate. The previous papers which reported the inhibitory effects of polyphenolic compounds on osteoclast differentiation and/or bone formation are not cited. The reviewer finds it very strange.

5. English is bad.

Minor points:

1. Careful English proofreading of the manuscript is prerequisite before submission to any journal.

2. All microscopic pictures (Fig. 1A, 2B, 3B, 5A, 5B, 6) lack scale bars.

3. Concentrations of the polyphenolic compounds used in the experiments is not indicated. In other papers, 10 μM is the concentration most often used and 50 μM is extremely high. In fact, administration of 50 μM compounds killed the cells (line 303, Fig. S2). The cytotoxicity of the compounds likely causes cell damages and suppresses osteoclast differentiation in this study and the manuscript only reports the fact that osteoclast differentiation of RAW264.7 cell was inhibited in the presence of high concentration TA, GA or EA. It is very likely that the cells were dying, as shown in Figure 6 and as the authors interpret (line 155, 550-553).

5. Unify the names, MCSF or M-CSF.

6. Unify the names, JC1 or JC-1. The manufacturer calls the dye JC-1.

7. Line 229, 243, “Mounted on slide glass with DAPI” is impossible.

8. Line 260-271, shold be moved to figure legends of Fig. S2.

9. Line 248-258, the protocols described in the manuscript do not measure intracellular calcium concentrations of the cells.

10. Line 378-380, the results in Figure 3B differ from the description in the text.

11. The results shown in Figure 3A and Figure 4B should be similar, but they are clearly different.

12. The resolution of Figure 5 is very low and the reviewer cannot evaluate it.

13. The authors must rewrite the Conclusions.

14. Reference 35 should be replaced with an appropriate literature(s).

15. Reference 42 should be replaced with an appropriate literature(s).

Author Response

Major points:

  1. The evidence presented in the manuscript dose not sound new.

We respectfully disagree with this comment. The autophagy and mitochondrial dysfunctions were not studied before.

  1. Not causal relationship but only parallelisms are presented. The authors have to present the evidence that RAW246.7 cells conduct autophagy and differentiate into osteoclast in the presence of the polyphenolic compounds when ROS are produced, and that RAW246.7 cells conduct autophagy and differentiate into osteoclast in the presence of the polyphenolic compounds when mitochondria are depolarized, to insist that the polyphenolic compounds prevent RAW246.71 cells from differentiating into osteoclasts by suppressing autophagy through loss of mitochondrial membrane potential and suppression of ROS production, as the authors titled the manuscript.

This is a great suggestion and we are planning to perform those experiments in our future project, which is beyond the scope of this manuscript.

  1. The cultured cells used in the study are probably under bad conditions, making the obtained results not-reliable. Perhaps, the concentrations of the polyphenolic compounds used was high and cytotoxic.

The concentrations used throughout the project (30 uM) were below toxic level, which was determined by the MTT assay. We have included MTT assay for testing the cell survivability, and included as a supplementary figure 2. in the manuscript.

  1. Citation is not appropriate. The previous papers which reported the inhibitory effects of polyphenolic compounds on osteoclast differentiation and/or bone formation are not cited. The reviewer finds it very strange.

We have included appropriate references in the revised version of the manuscript.

  1. English is bad.

We have thoroughly edited the English in the revised version of the manuscript.

Minor points:

  1. Careful English proofreading of the manuscript is prerequisite before submission to any journal.

We have thoroughly edited the English in the revised version of the manuscript.

  1. All microscopic pictures (Fig. 1A, 2B, 3B, 5A, 5B, 6) lack scale bars.

Scale bars are already present in Figures 1A, 2B, 5A, 5B and 6. To show the resolution and scale bar in the images, we have increased the size of the Figures and separated each panel of Fig.5 and made Fig.6 and 7. What was previously Fig. 6 became Fig. 8. We have included new scale bars in Fig. 3B, which were missing. 

  1. Concentrations of the polyphenolic compounds used in the experiments is not indicated. In other papers, 10 μM is the concentration most often used and 50 μM is extremely high. In fact, administration of 50 μM compounds killed the cells (line 303, Fig. S2). The cytotoxicity of the compounds likely causes cell damages and suppresses osteoclast differentiation in this study and the manuscript only reports the fact that osteoclast differentiation of RAW264.7 cell was inhibited in the presence of high concentration TA, GA or EA. It is very likely that the cells were dying, as shown in Figure 6 and as the authors interpret (line 155, 550-553).

We used 30 μM concentration of polyphenolic compound throughout the project, as is written in line 303, which might have been missed. It is evident from supplementary figure S2 that minimum cell death was observed within 48 hours using 30 μM conc. of polyphenolic compounds. In addition, Fig. 6 indicates that in OC differentiation medium in presence of EA, GA, and TA inhibited formation of OCs. We did not mention anything about the cell viability in the line 155, 550-553, it might be a mistake.

  1. Unify the names, MCSF or M-CSF.

We have unified M-CSF throughout the manuscript, and defined the abbreviation when mentioned first.

  1. Unify the names, JC1 or JC-1. The manufacturer calls the dye JC-1.

JC1 has been corrected and unified by writing it as JC-1 instead of JC1.

  1. Line 229, 243, “Mounted on slide glass with DAPI” is impossible.

We used DAPI in mounting solution (which is available commercially from Invitrogen, ThermoFisher, # P36931), as we mistakenly provided wrong catalog number in the manuscript and fixed that as well.

  1. Line 260-271, should be moved to figure legends of Fig. S2.

We have moved the lines 260-271 to the figure legends of Fig. S2.

  1. Line 248-258, the protocols described in the manuscript do not measure intracellular calcium concentrations of the cells.

We are sorry for the wrong method. We have corrected with the appropriate method in the revised manuscript.

  1. Line 378-380, the results in Figure 3B differ from the description in the text.

We have correctly mentioned about the autophagosome formation in the osteoclasts, which were detected by using MDC staining and addition of polyphenolic compounds reduces the formation of autophagosomes in the cells.

  1. The results shown in Figure 3A and Figure 4B should be similar, but they are clearly different.

As OC differentiation progressed, both genomic and protein expression of Becn1 and Atg7 increased significantly, whereas those values were decreased when TA, GA, and EA were added to the cells during differentiation process. We are now including the western blot measurement data as a histogram in supplementary figures 4-7.  

  1. The resolution of Figure 5 is very low, and the reviewer cannot evaluate it.

To enhance the resolution, we have increased the size of the images and also split the Fig. 5 into 3 (Fig5-7) in the revised version of the manuscript.

  1. The authors must rewrite the Conclusions.

We have rewritten the conclusion part in the revised manuscript.

  1. Reference 35 should be replaced with an appropriate literature(s).

We are sorry for the wrong reference. We have replaced the ref. 35 with appropriate literature.

  1. Reference 42 should be replaced with an appropriate literature(s).

We are sorry for the wrong reference. We have replaced the ref. 42 with appropriate literature.

Reviewer 2 Report

The manuscript by Laha et al. reports that polyphenolic compounds inhibit osteoclast differentiation by reducing autophagy through limiting ROS and the mitochondrial membrane potential. While the study is well designed and results obtained have a value, the manuscript is well written and easy to follow. Nonetheless, I have several minor concerns:  

- In my opinion, the results from the chapter 3.5 presented within Fig. 4 as westerns should be presented as histograms as a result of digitalization of three independent repeats.

- The quality of Fig. 5 within the version of the manuscript I have is extremely low.

- Abstract, Introduction, Discussion and Conclusions are lacking any mentions and discussions of the practical value for the goals of the study and results obtained.

Author Response

The manuscript by Laha et al. reports that polyphenolic compounds inhibit osteoclast differentiation by reducing autophagy through limiting ROS and the mitochondrial membrane potential. While the study is well designed and results obtained have a value, the manuscript is well written and easy to follow. Nonetheless, I have several minor concerns:  

- In my opinion, the results from the chapter 3.5 presented within Fig. 4 as westerns should be presented as histograms as a result of digitalization of three independent repeats.

We have quantified and analyzed the average three results of western blotting using Image J analysis and represented them graphically in the supplementary figure 4-7. 

- The quality of Fig. 5 within the version of the manuscript I have is extremely low.

To enhance the resolution, we have increased the size of the images and also split the Fig. 5 into 3 (Fig. 5-7) in the revised version of the manuscript.

- Abstract, Introduction, Discussion and Conclusions are lacking any mentions and discussions of the practical value for the goals of the study and results obtained.

In abstract, introduction, discussion, and conclusion we have mentioned that the polyphenolic compounds basically help in the prevention of OC differentiation through preventing mitochondrial functions through various pathways. 

Reviewer 3 Report

In the manuscript, the authors investigated the inhibition effects of several selected polyphenols on osteoclast differentiation and their active mechanisms. In general, the authors have completed a reasonable study with very informative data on the relative activities of the selected compounds, such as tannic acid (TA), gallic acid (GA) and ellagic acid (EA). Moreover, the hypothesis and objectives are sound and plainly exposed, methods are adequate and results are clearly shown and discussed. However, the major concerns are as followings.

1. There were no comparison studies among these three selected compounds from the statistical analyses.

2. To clarify the mechanism of action of the selected compounds on the osteoclast differentiation, a brief graph for presenting the supposed pathway might be better shown in the study to strengthen focus.

Author Response

In the manuscript, the authors investigated the inhibition effects of several selected polyphenols on osteoclast differentiation and their active mechanisms. In general, the authors have completed a reasonable study with very informative data on the relative activities of the selected compounds, such as tannic acid (TA), gallic acid (GA) and ellagic acid (EA). Moreover, the hypothesis and objectives are sound and plainly exposed, methods are adequate and results are clearly shown and discussed. However, the major concerns are as followings.

  1. There were no comparison studies among these three selected compounds from the statistical analyses.

Our study is not a comparative study, and we focused on investigating some of the polyphenolic compounds that prevented OC differentiation. In future, we plan to compare some of the polyphenolic compounds that are structurally different and would like to define which structure is responsible for the prevention of OC differentiation. However, those studies are beyond the scope of this manuscript.

  1. To clarify the mechanism of action of the selected compounds on the osteoclast differentiation, a brief graph for presenting the supposed pathway might be better shown in the study to strengthen focus.

We have included a new graphical abstract representing findings of our manuscript in our revised version of the manuscript.

Round 2

Reviewer 1 Report

The MS has been improved but still far from can be published.

Major points:

1. The evidence presented in the manuscript dose not sound new.

We respectfully disagree with this comment. The autophagy and mitochondrial dysfunctions were not studied before.

R:  Autophagy in osteoclast differentiation has been studied before: doi: 10.3390/biom10101398, doi.org/10.1016/j.bbrc.2020.04.155.

Mitochondria dysfunction in osteoclast differentiation has been studied before: doi: 10.1111/j.1749-6632.2009.05377.x, DOI:10.1007/s00281-019-00757-0.

The reviewer finds some studies which investigated inhibitory activities of polyphenol compounds in osteoclast differentiation: doi/full/10.1016/j.odw.2016.10.007, doi.org/10.1016/j.lfs.2018.06.013, doi.org/10.3945/cdn.117.000406, doi.org/10.1016/j.imr.2015.02.002, DOI: 10.1007/s12272-016-0790-0.

The reviewer finds it unnatural that the authors do not cite these references

2. Not causal relationship but only parallelisms are presented. The authors have to present the evidence that RAW246.7 cells conduct autophagy and differentiate into osteoclast in the presence of the polyphenolic compounds when ROS are produced, and that RAW246.7 cells conduct autophagy and differentiate into osteoclast in the presence of the polyphenolic compounds when mitochondria are depolarized, to insist that the polyphenolic compounds prevent RAW246.71 cells from differentiating into osteoclasts by suppressing autophagy through loss of mitochondrial membrane potential and suppression of ROS production, as the authors titled the manuscript.

This is a great suggestion and we are planning to perform those experiments in our future project, which is beyond the scope of this manuscript.

R: If the authors think so, the title of the MS should be changed.

3. The cultured cells used in the study are probably under bad conditions, making the obtained results not-reliable. Perhaps, the concentrations of the polyphenolic compounds used was high and cytotoxic.

The concentrations used throughout the project (30 uM) were below toxic level, which was determined by the MTT assay. We have included MTT assay for testing the cell survivability, and included as a supplementary figure 2. in the manuscript.

R:  We cannot conclude that there is no toxicity, just because cells do not die. Fig. 6 (Fig. 8 in the revised MS) strongly suggests that the cells were under the bad conditions, since a half of the cells administrated with 30 uM EA or GA lost mitochondrial membrane potential.

4. Citation is not appropriate. The previous papers which reported the inhibitory effects of polyphenolic compounds on osteoclast differentiation and/or bone formation are not cited. The reviewer finds it very strange.

We have included appropriate references in the revised version of the manuscript.

R: The reviewer finds that only two references (35 and 42), which the reviewer requests to replace, have been changed in the revised MS.

Autophagy in osteoclast differentiation has been studied before: doi: 10.3390/biom10101398, doi.org/10.1016/j.bbrc.2020.04.155.

Mitochondria dysfunction in osteoclast differentiation has been studied before: doi: 10.1111/j.1749-6632.2009.05377.x, DOI:10.1007/s00281-019-00757-0.

The reviewer finds some studies which investigated inhibitory activities of polyphenol compounds in osteoclast differentiation: doi/full/10.1016/j.odw.2016.10.007, doi.org/10.1016/j.lfs.2018.06.013, doi.org/10.3945/cdn.117.000406, doi.org/10.1016/j.imr.2015.02.002, DOI: 10.1007/s12272-016-0790-0.

The reviewer finds it unnatural that the authors do not cite these references

5. English is bad.

We have thoroughly edited the English in the revised version of the manuscript.

 R: The MS has been revised, although it still can be improved to be a MS written in good English: “autophagic genetic expression” should be “expression of autophagy-related genes” (line 356-357).

Minor points:

1. Careful English proofreading of the manuscript is prerequisite before submission to any journal.

We have thoroughly edited the English in the revised version of the manuscript.

 R: The MS has been revised, although it still can be improved to be a MS written in good English: “autophagic genetic expression” should be “expression of autophagy-related genes” (line 356-357).

2. All microscopic pictures (Fig. 1A, 2B, 3B, 5A, 5B, 6) lack scale bars.

Scale bars are already present in Figures 1A, 2B, 5A, 5B and 6. To show the resolution and scale bar in the images, we have increased the size of the Figures and separated each panel of Fig.5 and made Fig.6 and 7. What was previously Fig. 6 became Fig. 8. We have included new scale bars in Fig. 3B, which were missing. 

R:  The reviewer finds scale bars in Fig 1B but the length is not marked. The reviewer finds scale bars in Fig 2B but the length is not marked. The reviewer finds scale bars in Fig 3B but the length is not marked. The reviewer finds very small and useless scale bars in Fig 5 and Fig. 6 and the length is not marked. 

3. Concentrations of the polyphenolic compounds used in the experiments is not indicated. In other papers, 10 μM is the concentration most often used and 50 μM is extremely high. In fact, administration of 50 μM compounds killed the cells (line 303, Fig. S2). The cytotoxicity of the compounds likely causes cell damages and suppresses osteoclast differentiation in this study and the manuscript only reports the fact that osteoclast differentiation of RAW264.7 cell was inhibited in the presence of high concentration TA, GA or EA. It is very likely that the cells were dying, as shown in Figure 6 and as the authors interpret (line 155, 550-553).

We used 30 μM concentration of polyphenolic compound throughout the project, as is written in line 303, which might have been missed. It is evident from supplementary figure S2 that minimum cell death was observed within 48 hours using 30 μM conc. of polyphenolic compounds. In addition, Fig. 6 indicates that in OC differentiation medium in presence of EA, GA, and TA inhibited formation of OCs. We did not mention anything about the cell viability in the line 155, 550-553, it might be a mistake.

R:  The authors only described the concentration (30 uM) in the section 2.4 of the Materials and Methods (line 149) and not in the other sections of the Materials and Methods or in the figure legends. Readers can not determine the concentration used in each experiment even in the revised MS.

In line 303, the authors write that “We did not notice any remarkable cell death using the 30 μM concentration of EA, GA or TA (Supplementary Figure 2).”, and the reviewer can not understand this sentence that the authors used 30 μM concentration of polyphenolic compounds throughout the project. 

In Fig. 6 (Fig. 8 in the revised MS), the lost of red fluorescence of JC-1 in a half of the cells administrated with EA or GA indicates that mitochondria lost membrane potential in those cells. This indicates that the cells were going to dye or under very bad conditions.

5. Unify the names, MCSF or M-CSF.

We have unified M-CSF throughout the manuscript, and defined the abbreviation when mentioned first.

R: The reviewer has confirmed that it has been fixed.

6. Unify the names, JC1 or JC-1. The manufacturer calls the dye JC-1.

JC1 has been corrected and unified by writing it as JC-1 instead of JC1.

R: The reviewer has confirmed that it has been fixed.

7. Line 229, 243, “Mounted on slide glass with DAPI” is impossible.

We used DAPI in mounting solution (which is available commercially from Invitrogen, ThermoFisher, # P36931), as we mistakenly provided wrong catalog number in the manuscript and fixed that as well.

R: The reviewer understands that the authors mounted the specimens with DAPI, not using DAPI. The authors have to revise “were mounted on glass slides using DAPI” (line 236), since DAPI is a DNA-staining fluorescence dye and not a mounting medium. 

8. Line 260-271, should be moved to figure legends of Fig. S2.

We have moved the lines 260-271 to the figure legends of Fig. S2.

R: The reviewer has confirmed that it has been fixed.

9. Line 248-258, the protocols described in the manuscript do not measure intracellular calcium concentrations of the cells.

We are sorry for the wrong method. We have corrected with the appropriate method in the revised manuscript.

R:  The authors have to make clear what the authors measured in this experiment. Intracellular Ca2+ concentration and amount of Ca2+ in cell are different. “Intracellular [Ca2+]” in the title of y axis of Fig. 7 in the revised MS indicates concentration of Ca2+ in cytosol and generally it is 100-150 nM in live resting cells. The authors used a kit (ab102505) which measures the amount of Ca2+ in cell lysate. The authors have to revise the title of y-axis of Fig. 7 to “amount of intracellular Ca2+”. 

10. Line 378-380, the results in Figure 3B differ from the description in the text.

We have correctly mentioned about the autophagosome formation in the osteoclasts, which were detected by using MDC staining and addition of polyphenolic compounds reduces the formation of autophagosomes in the cells.

R: All of the cells contained vesicles those were brightly stained with MDC and the reviewer can not conclude the reduction of “autophagosome” in Fig. 3. 

11. The results shown in Figure 3A and Figure 4B should be similar, but they are clearly different.

As OC differentiation progressed, both genomic and protein expression of Becn1 and Atg7 increased significantly, whereas those values were decreased when TA, GA, and EA were added to the cells during differentiation process. We are now including the western blot measurement data as a histogram in supplementary figures 4-7.  

R: The densitometric result of GAPDH/LC3B Ratios which is presented in Fig. S5 and the western blotting image shown in Fig 4B do not look the same.

12. The resolution of Figure 5 is very low, and the reviewer cannot evaluate it.

To enhance the resolution, we have increased the size of the images and also split the Fig. 5 into 3 (Fig5-7) in the revised version of the manuscript.

R: It is necessary to digitalize and compare the fluorescence intensities shown in Fig. 5 to conclude that “ the level of ROS as determined by DCFDA staining (Fig. 5) and mitoSOX staining (Fig. 6), was remarkably decreased during differentiation (line 411-413), since the decrease in the DCFDA fluorescence shown in Fig. 5 is not clear. Scale bars are too fine and useless.

13. The authors must rewrite the Conclusions.

We have rewritten the conclusion part in the revised manuscript.

R: The reviewer has confirmed that it has been re-written and improved.

14. Reference 35 should be replaced with an appropriate literature(s).

We are sorry for the wrong reference. We have replaced the ref. 35 with appropriate literature.

R:  The reviewer has confirmed that it has been replaced. However, the sentence in the revised MS that “It was shown that the elevation of intercellular Ca2+ is due to a higher release of Ca2+ from the sarcoplasmic reticulum and a simultaneous influx of Ca2+ via voltage-gated Ca2+ channels [35]” describes the machinery of increase in intracellular Ca2+ concentration in muscle. This sentence should be rewritten for the Ca2+ increase in osteoclast. Now, the reviewer is sure that the authors confused concentration of intracellular Ca2+ and amount of Ca2+ in cell.

15. Reference 42 should be replaced with an appropriate literature(s).

We are sorry for the wrong reference. We have replaced the ref. 42 with appropriate literature.

R: The reviewer has confirmed that it has been fixed.

Additional requirement

1. Two authors are newly added in the revised MS but authors contributions has not been up-dated.

Author Response

Open Review

English language and style

( ) Extensive editing of English language and style required
( ) Moderate English changes required
(x) English language and style are fine/minor spell check required
( ) I don't feel qualified to judge about the English language and style

Yes

Can be improved

Must be improved

Not applicable

Does the introduction provide sufficient background and include all relevant references?

( )

( )

(x)

( )

Are all the cited references relevant to the research?

( )

( )

(x)

( )

Is the research design appropriate?

( )

( )

(x)

( )

Are the methods adequately described?

( )

(x)

( )

( )

Are the results clearly presented?

( )

( )

(x)

( )

Are the conclusions supported by the results?

( )

( )

(x)

( )

Comments and Suggestions for Authors

The MS has been improved but still far from can be published.

Major points:

  1. The evidence presented in the manuscript dose not sound new.

We respectfully disagree with this comment. The autophagy and mitochondrial dysfunctions were not studied before.

R:  Autophagy in osteoclast differentiation has been studied before: doi: 10.3390/biom10101398, doi.org/10.1016/j.bbrc.2020.04.155.

Mitochondria dysfunction in osteoclast differentiation has been studied before: doi: 10.1111/j.1749-6632.2009.05377.x, DOI:10.1007/s00281-019-00757-0.

The reviewer finds some studies which investigated inhibitory activities of polyphenol compounds in osteoclast differentiation: doi/full/10.1016/j.odw.2016.10.007, doi.org/10.1016/j.lfs.2018.06.013, doi.org/10.3945/cdn.117.000406, doi.org/10.1016/j.imr.2015.02.002, DOI: 10.1007/s12272-016-0790-0.

The reviewer finds it unnatural that the authors do not cite these references

We have included all the citations (Ref.#38-45) in the revised version of the manuscript (line 507-508) to give more context to the study.

  1. Not causal relationship but only parallelisms are presented. The authors have to present the evidence that RAW246.7 cells conduct autophagy and differentiate into osteoclast in the presence of the polyphenolic compounds when ROS are produced, and that RAW246.7 cells conduct autophagy and differentiate into osteoclast in the presence of the polyphenolic compounds when mitochondria are depolarized, to insist that the polyphenolic compounds prevent RAW246.71 cells from differentiating into osteoclasts by suppressing autophagy through loss of mitochondrial membrane potential and suppression of ROS production, as the authors titled the manuscript.

This is a great suggestion and we are planning to perform those experiments in our future project, which is beyond the scope of this manuscript.

R: If the authors think so, the title of the MS should be changed.

We have changed the title that does not link the osteoclastogenesis and other observations as strongly as before.

  1. The cultured cells used in the study are probably under bad conditions, making the obtained results not-reliable. Perhaps, the concentrations of the polyphenolic compounds used was high and cytotoxic.

The concentrations used throughout the project (30 uM) were below toxic level, which was determined by the MTT assay. We have included MTT assay for testing the cell survivability, and included as a supplementary figure 2. in the manuscript.

R:  We cannot conclude that there is no toxicity, just because cells do not die. Fig. 6 (Fig. 8 in the revised MS) strongly suggests that the cells were under the bad conditions, since a half of the cells administrated with 30 uM EA or GA lost mitochondrial membrane potential.

We strongly believe that the toxicity is not the issue here as we can harvest RNA and protein from those cells after the experiment and evaluate those.

  1. Citation is not appropriate. The previous papers which reported the inhibitory effects of polyphenolic compounds on osteoclast differentiation and/or bone formation are not cited. The reviewer finds it very strange.

We have included appropriate references in the revised version of the manuscript.

R: The reviewer finds that only two references (35 and 42), which the reviewer requests to replace, have been changed in the revised MS.

Autophagy in osteoclast differentiation has been studied before: doi: 10.3390/biom10101398, doi.org/10.1016/j.bbrc.2020.04.155.

Mitochondria dysfunction in osteoclast differentiation has been studied before: doi: 10.1111/j.1749-6632.2009.05377.x, DOI:10.1007/s00281-019-00757-0.

The reviewer finds some studies which investigated inhibitory activities of polyphenol compounds in osteoclast differentiation: doi/full/10.1016/j.odw.2016.10.007, doi.org/10.1016/j.lfs.2018.06.013, doi.org/10.3945/cdn.117.000406, doi.org/10.1016/j.imr.2015.02.002, DOI: 10.1007/s12272-016-0790-0.

The reviewer finds it unnatural that the authors do not cite these references

We have included the citations in the revised version of the manuscript to give more context to the study.

  1. English is bad.

We have thoroughly edited the English in the revised version of the manuscript.

 R: The MS has been revised, although it still can be improved to be a MS written in good English: “autophagic genetic expression” should be “expression of autophagy-related genes” (line 356-357).

 We have corrected the text accordingly.

Minor points:

  1. Careful English proofreading of the manuscript is prerequisite before submission to any journal.

We have thoroughly edited the English in the revised version of the manuscript.

 R: The MS has been revised, although it still can be improved to be a MS written in good English: “autophagic genetic expression” should be “expression of autophagy-related genes” (line 356-357).

We have corrected the text accordingly.

  1. All microscopic pictures (Fig. 1A, 2B, 3B, 5A, 5B, 6) lack scale bars.

Scale bars are already present in Figures 1A, 2B, 5A, 5B and 6. To show the resolution and scale bar in the images, we have increased the size of the Figures and separated each panel of Fig.5 and made Fig.6 and 7. What was previously Fig. 6 became Fig. 8. We have included new scale bars in Fig. 3B, which were missing. 

R:  The reviewer finds scale bars in Fig 1B but the length is not marked. The reviewer finds scale bars in Fig 2B but the length is not marked. The reviewer finds scale bars in Fig 3B but the length is not marked. The reviewer finds very small and useless scale bars in Fig 5 and Fig. 6 and the length is not marked. 

We have added the length of the scale bar in the legends of the Figures those were missing the length. 

  1. Concentrations of the polyphenolic compounds used in the experiments is not indicated. In other papers, 10 μM is the concentration most often used and 50 μM is extremely high. In fact, administration of 50 μM compounds killed the cells (line 303, Fig. S2). The cytotoxicity of the compounds likely causes cell damages and suppresses osteoclast differentiation in this study and the manuscript only reports the fact that osteoclast differentiation of RAW264.7 cell was inhibited in the presence of high concentration TA, GA or EA. It is very likely that the cells were dying, as shown in Figure 6 and as the authors interpret (line 155, 550-553).

We used 30 μM concentration of polyphenolic compound throughout the project, as is written in line 303, which might have been missed. It is evident from supplementary figure S2 that minimum cell death was observed within 48 hours using 30 μM conc. of polyphenolic compounds. In addition, Fig. 6 indicates that in OC differentiation medium in presence of EA, GA, and TA inhibited formation of OCs. We did not mention anything about the cell viability in the line 155, 550-553, it might be a mistake.

R:  The authors only described the concentration (30 uM) in the section 2.4 of the Materials and Methods (line 149) and not in the other sections of the Materials and Methods or in the figure legends. Readers can not determine the concentration used in each experiment even in the revised MS.

We have added concentrations of the polyphenolic compound in each of the experiments.

In line 303, the authors write that “We did not notice any remarkable cell death using the 30 μM concentration of EA, GA or TA (Supplementary Figure 2).”, and the reviewer can not understand this sentence that the authors used 30 μM concentration of polyphenolic compounds throughout the project. 

In Fig. 6 (Fig. 8 in the revised MS), the lost of red fluorescence of JC-1 in a half of the cells administrated with EA or GA indicates that mitochondria lost membrane potential in those cells. This indicates that the cells were going to dye or under very bad conditions.

We strongly believe that the toxicity is not the issue here as we can harvest RNA and protein from those cells after the experiment and evaluate those. So, we sincerely disagree with the comment made by the reviewer.

  1. Unify the names, MCSF or M-CSF.

We have unified M-CSF throughout the manuscript, and defined the abbreviation when mentioned first.

R: The reviewer has confirmed that it has been fixed.

  1. Unify the names, JC1 or JC-1. The manufacturer calls the dye JC-1.

JC1 has been corrected and unified by writing it as JC-1 instead of JC1.

R: The reviewer has confirmed that it has been fixed.

  1. Line 229, 243, “Mounted on slide glass with DAPI” is impossible.

We used DAPI in mounting solution (which is available commercially from Invitrogen, ThermoFisher, # P36931), as we mistakenly provided wrong catalog number in the manuscript and fixed that as well.

R: The reviewer understands that the authors mounted the specimens with DAPI, not using DAPI. The authors have to revise “were mounted on glass slides using DAPI” (line 236), since DAPI is a DNA-staining fluorescence dye and not a mounting medium. 

We have corrected the errors.

  1. Line 260-271, should be moved to figure legends of Fig. S2.

We have moved the lines 260-271 to the figure legends of Fig. S2.

R: The reviewer has confirmed that it has been fixed.

  1. Line 248-258, the protocols described in the manuscript do not measure intracellular calcium concentrations of the cells.

We are sorry for the wrong method. We have corrected with the appropriate method in the revised manuscript.

R:  The authors have to make clear what the authors measured in this experiment. Intracellular Ca2+ concentration and amount of Ca2+ in cell are different. “Intracellular [Ca2+]” in the title of y axis of Fig. 7 in the revised MS indicates concentration of Ca2+ in cytosol and generally it is 100-150 nM in live resting cells. The authors used a kit (ab102505) which measures the amount of Ca2+ in cell lysate. The authors have to revise the title of y-axis of Fig. 7 to “amount of intracellular Ca2+”.

We have fixed the Title of the Y-Axis of the Fig 7. 

  1. Line 378-380, the results in Figure 3B differ from the description in the text.

We have correctly mentioned about the autophagosome formation in the osteoclasts, which were detected by using MDC staining and addition of polyphenolic compounds reduces the formation of autophagosomes in the cells.

R: All of the cells contained vesicles those were brightly stained with MDC and the reviewer can not conclude the reduction of “autophagosome” in Fig. 3.

We have corrected our overstatement of the result. 

  1. The results shown in Figure 3A and Figure 4B should be similar, but they are clearly different.

As OC differentiation progressed, both genomic and protein expression of Becn1 and Atg7 increased significantly, whereas those values were decreased when TA, GA, and EA were added to the cells during differentiation process. We are now including the western blot measurement data as a histogram in supplementary figures 4-7.  

R: The densitometric result of GAPDH/LC3B Ratios which is presented in Fig. S5 and the western blotting image shown in Fig 4B do not look the same.

We have considered the lower band of the LC3B, and that is represented correctly in the supplementary Figure S5.

  1. The resolution of Figure 5 is very low, and the reviewer cannot evaluate it.

To enhance the resolution, we have increased the size of the images and also split the Fig. 5 into 3 (Fig5-7) in the revised version of the manuscript.

R: It is necessary to digitalize and compare the fluorescence intensities shown in Fig. 5 to conclude that “ the level of ROS as determined by DCFDA staining (Fig. 5) and mitoSOX staining (Fig. 6), was remarkably decreased during differentiation (line 411-413), since the decrease in the DCFDA fluorescence shown in Fig. 5 is not clear. Scale bars are too fine and useless.

We have corrected our statement, and added the length of the scale bar in the legends of the Figures that were missing the length. 

  1. The authors must rewrite the Conclusions.

We have rewritten the conclusion part in the revised manuscript.

R: The reviewer has confirmed that it has been re-written and improved.

  1. Reference 35 should be replaced with an appropriate literature(s).

We are sorry for the wrong reference. We have replaced the ref. 35 with appropriate literature.

R:  The reviewer has confirmed that it has been replaced. However, the sentence in the revised MS that “It was shown that the elevation of intercellular Ca2+ is due to a higher release of Ca2+ from the sarcoplasmic reticulum and a simultaneous influx of Ca2+ via voltage-gated Ca2+ channels [35]” describes the machinery of increase in intracellular Ca2+ concentration in muscle. This sentence should be rewritten for the Ca2+ increase in osteoclast. Now, the reviewer is sure that the authors confused concentration of intracellular Ca2+ and amount of Ca2+ in cell.

Sorry for our mistake. Thank you for your clarification. We have corrected the sentence accordingly.

  1. Reference 42 should be replaced with an appropriate literature(s).

We are sorry for the wrong reference. We have replaced the ref. 42 with appropriate literature.

R: The reviewer has confirmed that it has been fixed.

Additional requirement

  1. Two authors are newly added in the revised MS but authors contributions has not been up-dated.

We have corrected the errors.

Submission Date

02 June 2022

Date of this review

16 Aug 2022 02:18:08